# Zinc Differentially Modulates the Assembly of Soluble and Polymerized Vimentin

**DOI:** 10.3390/ijms21072426

**Published:** 2020-03-31

**Authors:** Andreia Mónico, Silvia Zorrilla, Germán Rivas, Dolores Pérez-Sala

**Affiliations:** Department of Structural and Chemical Biology, Centro de Investigaciones Biológicas Margarita Salas, Ramiro de Maeztu, 9, 28040 Madrid, Spain; andreiamonico89@gmail.com (A.M.); silvia@cib.csic.es (S.Z.); grivas@cib.csic.es (G.R.)

**Keywords:** vimentin, zinc, cysteine, redox sensing, intermediate filaments, cysteine mutant, filament bundling, filament width, divalent cations, magnesium

## Abstract

The intermediate filament protein vimentin constitutes a critical sensor for electrophilic and oxidative stress. We previously showed that vimentin interacts with zinc, which affects its assembly and redox sensing. Here, we used vimentin wt and C328S, an oxidation-resistant mutant showing improved NaCl-induced polymerization, to assess the impact of zinc on soluble and polymerized vimentin by light scattering and electron microscopy. Zinc acts as a switch, reversibly inducing the formation of vimentin oligomeric species. High zinc concentrations elicit optically-detectable vimentin structures with a characteristic morphology depending on the support. These effects also occur in vimentin C328S, but are not mimicked by magnesium. Treatment of vimentin with micromolar ZnCl_2_ induces fibril-like particles that do not assemble into filaments, but form aggregates upon subsequent addition of NaCl. In contrast, when added to NaCl-polymerized vimentin, zinc increases the diameter or induces lateral association of vimentin wt filaments. Remarkably, these effects are absent or attenuated in vimentin C328S filaments. Therefore, the zinc-vimentin interaction depends on the chemical environment and on the assembly state of the protein, leading to atypical polymerization of soluble vimentin, likely through electrostatic interactions, or to broadening and lateral association of preformed filaments through mechanisms requiring the cysteine residue. Thus, the impact of zinc on vimentin assembly and redox regulation is envisaged.

## 1. Introduction

Intermediate filaments play a fundamental role in the integration of cytoskeletal functions, and consequently in cell dynamics and behavior. The type III intermediate filament protein vimentin constitutes a key cytoskeletal element of mesenchymal cells. Vimentin plays key roles in essential cell functions such as division and migration and contributes to cellular structural support and plasticity and organelle positioning. In addition, recent reports have unveiled its complex interplay with other cytoskeletal systems [1,2] and its involvement in redox sensing [3,4], regulation of gene expression, [5] or protection of the nucleus and DNA from damage [6]. Moreover, biophysical techniques are providing high resolution information on the mechanics and dynamic performance of the vimentin network, with a clear impact on the physical properties of cells [6,7]. In pathophysiology, vimentin is a key marker and agent of epithelial mesenchymal transition and tumor malignancy [2,8,9,10] and is involved in bacterial and viral infections [11] and autoimmune diseases [12].

Vimentin’s structure is comprised of intrinsically disordered N-terminal (head) and C-terminal (tail) domains and a central rod segment of a coiled-coil structure. In vitro, vimentin monomers spontaneously form dimers and tetramers when purified from solutions containing chaotropic agents into low ionic strength buffers. In turn, these oligomers rapidly form filaments upon raising the ionic strength. Numerous excellent works have delineated the process of vimentin filament assembly starting from the association of parallel dimers into staggered antiparallel tetramers, and of these into “unit length filaments” (ULF), several of which engage end-to-end to form filaments that subsequently undergo a radial compaction process to yield mature filaments [13,14,15].

In cells, vimentin is subjected to an exquisite regulation, both by transcriptional and posttranscriptional mechanisms relying on a plethora of posttranslational modifications, including phosphorylation, glycosylation, SUMOylation, as well as non-enzymatic processes, like oxidative modifications [4,16]. In particular, modifications of the vimentin single cysteine residue, C328 in human vimentin, play a key role in filament assembly in vitro and network remodeling in cells [3,4,17]. Oxidative modifications of C328 disrupt in vitro filament formation in a manner dependent on the structure of the modifying moiety [4]. Moreover, C328 appears to occupy a key position in assembled filaments in such a way that modifications or substitutions of this residue by other amino acids affect vimentin performance in cellular functions. Thus, C328S vimentin is less efficient than the wild-type (wt) protein at forming extended networks and supporting organelle positioning in cells [3]. Moreover, cysteine-deficient vimentin displays an attenuated response against several types of stress [3,4]. Although knowledge of the regulation of vimentin is rapidly expanding, there are still several intriguing aspects like its potential interaction with regulatory or associated proteins and the mechanisms controlling assembly in the cellular context.

Given its amino acid composition, vimentin is known to behave as a polyelectrolyte and to interact with cations through charge-mediated interactions [18]. The interaction of vimentin with divalent cations, mainly magnesium and calcium, has long been known and characterized in detail in vitro [19,20,21]. Both magnesium and calcium have been shown to induce vimentin polymerization on their own when used at millimolar concentrations [20,22,23] and to increase the stiffness of vimentin networks, thereby acting as crosslinkers. Indeed, divalent cations were found to interact with the tail domain of vimentin, in particular with the last 11 amino acids, inducing its crosslinking [19]. In particular, Mg^2+^ ions appear to accumulate in the tail segments due to the abundance of negatively charged amino acids in this domain [24]. In microfluidic drops, concentrations of magnesium in the millimolar range augmented the compaction of preassembled (KCl-induced) vimentin filament networks until they became completely aggregated [25,26]. In cells, the role of divalent cations on vimentin structure and function can be highly complex. Besides a direct interaction with the protein, divalent cations could act on vimentin dynamics through multiple mechanisms, including regulation of proteases or kinases in the case of calcium [27,28,29], binding to ATP for kinase reactions or facilitating GTP binding to GTPases in the case of magnesium.

Zinc is the transition metal that most frequently acts as a cofactor of proteins [30]. Zinc interaction with proteins may play structural and catalytic roles or serve for its storage and transport in cells. Importantly, zinc availability has clear pathophysiological implications in diseases related to protein aggregation, oxidative stress, and immune function [3]. Zinc can interact with proteins with beneficial or deleterious consequences. Thus, zinc can provide interactions promoting protein stability, but also facilitates the formation of pathogenic aggregates, such as Aβ amyloid precipitates or Tau fibrils [31,32]. On the other hand, zinc can induce cellular effects by modulating the activity of transcription factors [33] and phosphatases [34]. Moreover, zinc can influence protein and cellular redox status due to its potential pro-antioxidant or pro-oxidant roles [35,36]. Therefore, the interactions between zinc and vimentin in vitro and in cells can be complex, and their delineation requires careful assessment.

The interaction of intermediate filament proteins with zinc was first noted for keratins, in which low millimolar zinc induced protein aggregation, elicited the reversible association of two to four keratin filaments, and promoted the formation of macrofilaments [37,38]. While studying the importance of the single cysteine residue of vimentin, we obtained evidence indicating that zinc could interact with vimentin in vitro, protecting this residue from alkylation [3]. Moreover, zinc promoted vimentin polymerization into insoluble structures per se. In cells, we observed that zinc availability contributed to network robustness and protection from oxidants [3]. Our observations suggested a supra-stoichiometric interaction of vimentin with zinc with preference over other divalent cations such as Mg^2+^. This led us to propose a role for the zinc-vimentin interaction in both vimentin and zinc homeostasis [39,40]. 

Nevertheless, the structural basis of the vimentin-zinc interaction is not known. Here, we studied the effect of zinc on vimentin assembly in vitro. Our results indicated that low micromolar zinc induced a differential modulation of the assembly in soluble vimentin and preformed filaments, which could have implications in physiological or pathophysiological processes.

## 2. Results

### 2.1. EDTA-Free Vimentin Reveals Differential Polymerization Features Derived from the Presence of the Cysteine Residue

Most protocols for purification and refolding of vimentin employ millimolar concentrations of EDTA as the stabilizing agent and metalloprotease inhibitor. Therefore, in order to study the effect of micromolar zinc on vimentin assembly, we employed a procedure for efficient removal of EDTA from vimentin preparations [41]. This procedure, which involves ultrafiltration and dialysis, yielded vimentin as a single band in SDS gels, even under non-reducing conditions, indicating that the protein is recovered in its reduced form (Figure 1A). This is important since we have previously shown that vimentin oxidation can hamper filament formation [4]. The functionality of the resulting protein preparation was validated by polymerization assays. Both vimentin wt and its C328S mutant were recovered in the pellet fraction from ultracentrifugation after addition of 150 mM NaCl, indicative of polymerization (Figure 1B). Next, the morphology of the structures present and the kinetics of polymerization were assessed by electron microscopy (EM). In the absence of NaCl, vimentin preparations appeared as homogeneous lattices of rod-like structures of 2–5 nm in width, generally shorter than 50 nm (Figure 1C), which likely correspond to the soluble oligomeric vimentin forms previously characterized [42]. Upon addition of NaCl, the formation of typical vimentin filaments was observed from the first time-point assessed, i.e., 5 min, at which, filament width was approximately 14 nm (Figure 1D). At later polymerization times, filaments narrowed gradually, in accordance with the radial compaction phenomenon (quantitated in Figure 1E). Interestingly, vimentin wt filaments were consistently wider than vimentin C328S filaments, reaching 12.3 and 11.8 nm in diameter, respectively (Figure 1E). In turn, vimentin C328S filaments were significantly longer than wt at early polymerization times (Figure 1F), suggesting a faster elongation and highlighting a role of the residue at position 328 in the assembly process. In summary, these virtually EDTA-free vimentin preparations were considered functionally appropriate to study the effect of zinc on vimentin oligomerization. Moreover, they allowed the observation of differences in the polymerization behavior of wt and C328S vimentin.

### 2.2. Micromolar Zinc Reversibly induces Vimentin Oligomerization

The effect of zinc on vimentin polymerization was studied first by ultracentrifugation assays (Figure 2A). Analysis of the supernatant and pellet fractions containing soluble and insoluble-polymerized vimentin, respectively, showed that micromolar ZnCl_2_ induced the polymerization of both vimentin wt and C328S, in such a way that in the presence of 250 µM ZnCl_2_, most of the vimentin was in the pellet. Although informative, centrifugation assays require long processing times. In order to study the effect of zinc on vimentin polymerization in more detail, we employed turbidity assays, which allow monitoring zinc-induced changes in real time (Figure 2B). These showed a steady increase in turbidity with increasing zinc concentrations for both vimentin wt and C328S, which reflects the formation of larger vimentin species already at concentrations of ZnCl_2_ in the micromolar range. An increase in turbidity did not saturate at the highest concentration used (300 µM). Nevertheless, higher concentrations or longer turbidity measurements were not considered due to decreases in absorbance compatible with sedimentation of large protein assemblies. 

Next, fluorescence anisotropy was employed to confirm the formation of oligomeric assemblies in the presence of micromolar zinc. Mixtures of unlabeled vimentin wt or C328S and the corresponding construct labeled with Alexa 488, as a tracer, were incubated with increasing concentrations of ZnCl_2_, and fluorescence anisotropy was measured (Figure 2C). A concentration-dependent increase in fluorescence anisotropy, compatible with the formation of larger protein species, was clearly observed with concentrations of ZnCl_2_ below 50 µM and corroborated the ability of micromolar zinc to induce vimentin oligomerization. Similar to the turbidity measurements, no significant differences in zinc-induced aggregation between vimentin wt and C328S were observed. Turbidity measurements were used to explore the dynamic nature of zinc-induced vimentin oligomerization. As shown in Figure 2D, incubation of vimentin with 150 µM ZnCl_2_ elicited a nearly immediate increase in turbidity that was stable during the time of observation. The addition of an equivalent concentration of the metal chelators, TPEN or EDTA, which form a 1:1 complex with zinc, immediately caused an abrupt decrease in turbidity, compatible with disassembly of zinc-induced oligomers. This decrease was more marked in the case of TPEN, which reduced turbidity down to basal levels. This could be related to the higher affinity of TPEN for zinc, compared to EDTA (K_d_ of 6.4 × 10^−16^ M and 6 × 10^−14^ M, for TPEN and EDTA, respectively) [43]. Interestingly, subsequent re-addition of 150 µM ZnCl_2_, final concentration, rapidly increased turbidity again up to levels similar to those attained after the initial addition of ZnCl_2_ in the presence of TPEN, but apparently higher in the presence of EDTA. This may reflect a lower chelating power of EDTA or additional effects on vimentin assembly [42] that call for caution in the use of this chelator. Nevertheless, these observations clearly show that the effect of zinc on vimentin oligomerization is reversible and suggest that micromolar zinc can act as a switch eliciting the formation of vimentin oligomeric species. Moreover, these results parallel observations in cells showing the cyclic disassembly and re-assembly of GFP-vimentin constructs upon addition and withdrawal of TPEN [3].

Importantly, the effect of zinc could not be mimicked by similar concentrations of another divalent cation, namely, Mg^2+^. As shown in Figure 3A, adding MgCl_2_ up to 500 µM did not increase the proportion of insoluble vimentin for any of the constructs. Turbidity assays demonstrated that millimolar concentrations of MgCl_2_ were required to elicit increases in absorbance similar to those induced by micromolar zinc (Figure 3B). For instance, a comparable increment in turbidity was observed with 150 µM ZnCl_2_ and 8 mM MgCl_2_, that is a 50-fold higher concentration of magnesium. These observations indicated that the mode of binding of the two divalent cations to the protein should be different, which could potentially result in distinct structural consequences.

Finally, we assessed the impact of several NaCl concentrations on vimentin turbidity and found that millimolar NaCl only marginally increased turbidity under the conditions of this assay (Figure 3C). 

### 2.3. The Morphology of Zinc-Induced Vimentin Assemblies Depends on the Support

Turbidity assays provide a quantitative measurement of vimentin oligomerization. However, they do not provide information on the morphology of the assemblies formed. Therefore, we employed several microscopy techniques to assess this point (Figure 4). When a solution of vimentin was incubated with increasing concentrations of ZnCl_2_ in a flat-bottomed plastic microplate, vimentin formed assemblies with a garland-like appearance that were detectable after 10 to 20 min of incubation with 200 µM ZnCl_2_ and became more abundant with higher zinc concentrations (Figure 4A). 

The formation of assemblies in a confined environment was also monitored by using a mixture of unlabeled vimentin wt and vimentin wt labeled with Alexa 488 (4:1), encapsulated in picoliter-sized aqueous droplets stabilized by lipids in an oil phase, and observed by fluorescence microscopy (Figure 4B). In the absence of zinc, vimentin was homogenously dispersed inside the droplets. Low concentrations of ZnCl_2_, e.g., 20 µM, did not alter this pattern detectably. However, when encapsulated in the presence of 100 to 200 µM concentrations of ZnCl_2_, polymorphic aggregates and occasional filamentous structures could be observed, causing an obvious redistribution of fluorescence (Figure 4B).

Consistent with our previous observations, when mixed with ZnCl_2_ on glass coverslips, vimentin immediately formed aggregates and arrays or fibers, the robustness of which increased with zinc concentration (Figure 4C). Moreover, these fibers could be stained with zinquin, a fluorescent probe that detects both free and protein-bound zinc. Zinc also induced the formation of fibers of vimentin C328S (Figure 4C). Conversely, incubation of vimentin on glass with equivalent concentrations of MgCl_2_ elicited some amorphous aggregates, but no fibers were detectable by optical microscopy, indicating that the zinc-vimentin interaction possesses specific features (Figure 4C). 

Therefore, the morphology of the zinc-induced vimentin assemblies formed appears to depend on the support used and the environmental conditions.

### 2.4. Assessment of Zinc-Induced Vimentin Structures by Electron Microscopy

Optical microscopy provides information on the morphology of the larger assemblies of vimentin. In order to assess the effects of zinc on smaller assemblies, we employed EM (Figure 5). As described above, vimentin in hypotonic buffer appeared mainly as weakly-defined small rod-like structures of 2–5 nm in width and 40–50 nm in apparent length (Figure 5A). Upon incubation with 10 µM ZnCl_2_, vimentin formed more easily distinguishable assemblies, consisting of rod-shaped particles of 9.70 ± 0.03 nm and 9.43 ± 0.03 nm in width, in vimentin wt and C328S, respectively, which coexisted with some non-assembled material and occasional larger aggregates. Moreover, 100 μM ZnCl_2_ further increased the diameter of vimentin structures up to 10.6 -11.5 nm, as depicted in Figure 5B. Remarkably, the length of zinc-induced structures did not suffer a noticeable variation with increasing zinc concentrations over the time of the experiment (1 h) (Figure 5C). Thus, these observations indicated that zinc promoted lateral association of vimentin oligomers into fibrils. Nevertheless, on average, these structures did not reach the dimensions typical of ULF, which display a characteristic length of approximately 60–65 nm and 16–20 nm in width [13,44]. Moreover, elongation or end-to end annealing of these fibrils did not appear to take place in a significant proportion, since the structures observed did not reach the length corresponding to two typical ULF during the time frame of the experiment (length was always below 100 nm).

### 2.5. Effect of Zinc on NaCl-Induced Polymerization and on Preformed Filaments

Vimentin polymerization in cells occurs in an isotonic medium, which, in vitro, is usually mimicked by adding millimolar concentrations of monovalent salts, i.e., NaCl or KCl. Therefore, turbidity assays were used to explore the effect of sequential addition of NaCl and ZnCl_2_ in different orders (Figure 6A). The addition of 150 µM ZnCl_2_ to vimentin rapidly increased turbidity. Subsequent addition of 150 mM NaCl caused a marked and immediate increase in turbidity, suggesting the formation of larger assemblies (Figure 6A, left panel). Conversely, the addition of ZnCl_2_ to vimentin after incubation with NaCl induced a more moderate increase in turbidity (Figure 6A, right panel). Then, the nature of the assemblies formed was assessed by electron microscopy. Strikingly, preincubation of vimentin with zinc precluded the formation of filaments upon the addition of NaCl and led to irregular assemblies that sometimes tended to form longitudinal arrays, and others appeared as tangled, non-linear structures (Figure 6B). This may indicate that the fibrils formed in the presence of zinc alone are not valid “starter units”; that is, they do not possess the proper alignment or conformation to undergo end-to-end annealing and compaction. It is noteworthy that the disrupting effect of zinc on subsequent NaCl-elicited elongation occurred with concentrations as low as 10 µM ZnCl_2_; that is, at a 2:1 zinc:vimentin ratio.

In contrast, when NaCl-induced preformed vimentin filaments were incubated in the presence of zinc, filament integrity was not altered (Figure 6C). Nevertheless, a clear increase in filament width, from 12.6 to 14.3 nm, was observed with the lowest ZnCl_2_ concentration assayed (10 µM). Higher concentrations of ZnCl_2_, up to 100 µM, did not induce further increases in filament diameter (Figure 6C). Importantly, this effect was observed only in vimentin wt filaments, since vimentin C328S filaments remained unaltered in the presence of zinc (quantitated in Figure 6D). Moreover, 100 µM ZnCl_2_ affected preformed filaments by inducing lateral association, preferentially in vimentin wt. Thus, in vimentin wt, bundles contained more filaments on average (3.4 ± 1.8 in wt vs. 2.8 ± 1.4 in C328S vimentin, average ± SD of over 200 bundles, *p* < 0.0001). Moreover, vimentin wt bundles were tightly packed, with a poor definition of individual filaments. In contrast, vimentin C328S filaments exposed to 100 µM ZnCl_2_ most frequently associated laterally in groups of two or three filaments that preserved their individual contours (Figure 6C). The distribution of filaments in bundles in vimentin wt and C328S is represented in Figure 6E, showing that vimentin wt appears more frequently than vimentin C328S in groups of three or more filaments, whereas the mutant appears more frequently than the wt in groups of one or two filaments.

## 3. Discussion

The regulation of vimentin filament assembly is not completely understood. Here, we showed that micromolar zinc exerted unique effects on the assembly of vimentin in vitro. Zinc induced the oligomerization of soluble vimentin into fibril-like structures apparently not competent for filament formation. However, in preformed filaments, it induced morphological changes consisting of filament widening and lateral association. These effects could have physiological and pathophysiological implications, as discussed below.

Importantly, the effects of zinc occurred at micromolar concentrations. Therefore, their characterization required minimizing the presence of metal chelators. Indeed, metal chelators such as EDTA or EGTA were previously shown to affect vimentin polymerization, reportedly by altering the ionic strength of the solution [42]. In addition, EDTA could interfere with vimentin assembly through other mechanisms, such as the occurrence of pH changes in solutions containing EDTA upon the addition of divalent cations [45]. Therefore, throughout this study, we used a virtually EDTA-free vimentin preparation [41].

In previous studies, we observed that zinc protected the vimentin single cysteine residue, C328, from alkylation [3], which has been proposed to be an indication of zinc binding [46]. Therefore, here, we explored the effect of zinc on both vimentin wt and a cysteine to serine mutant, C328S. The substitution of a cysteine residue by serine is considered the most conservative one regarding size and atomic composition. Although the single cysteine residue appears not to be essential for in vitro vimentin assembly [17], here, we observed subtle differences between the polymerization behavior of vimentin wt and C328S in vitro, consistent with a faster elongation and formation of narrower filaments in the case of the mutant. Interestingly, a vimentin C328S mutant has proven to be less efficient than vimentin wt for initial network formation and undergoes faster dynamics in cells [3], indicating that, functionally, serine cannot fully substitute for cysteine in vimentin. These observations, together with the evidence of the impact of cysteine oxidative modifications in filament formation [4], could imply that C328 is located at a position important for filament assembly.

Zinc-induced vimentin oligomerization is fully reversible, as suggested by previous observations [3], and demonstrated here by real-time monitoring of light scattering. The dimensions of zinc-induced vimentin fibrils appear to be different from those of tetramers or typical ULF. This could be due to a higher degree of compaction of tetramers or to an altered alignment of dimers or tetramers, not appropriate for ULF formation and elongation. Notably, the effect of zinc on the aggregation of soluble vimentin appears to be similar in the wt and in the C328S mutant. In this context, it should be taken into account that, although histidine and cysteine residues are the preferred ligands, serine residues can also participate in interactions with zinc [30]. Moreover, carboxylic amino acids, glutamate and aspartate, also provide sites for zinc interaction [35]. Thus, the presence of numerous carboxylic amino acids in the vimentin sequence could explain the ability of zinc to promote the aggregation of both wt and C328S mutant vimentin.

Remarkably, concentrations of zinc in the hundreds of micromolar or low millimolar range induce the appearance of optically detectable vimentin aggregates and fibers. This could be related to the polyelectrolyte properties of vimentin, as multivalent counterions, including Ca^2+^, Mg^2+^, and Mn^2+^ at concentrations of 10 mM, have been shown to induce bundling of proteins with negative charge densities at their surfaces, including actin and vimentin [18]. Moreover, counterions can induce the formation of polyelectrolyte protein lattices through crosslinking [18].

However, we observed that the effects of zinc on vimentin oligomerization could not be mimicked by similar concentrations of magnesium, highlighting the differences in the interaction of the two cations with vimentin. On the one hand, the concentrations of magnesium required to induce vimentin oligomerization, as monitored by light scattering, are approximately 50-fold higher than those of zinc. On the other hand, whereas zinc-induced oligomers appear to be elongation-incompetent units, magnesium has been shown to induce ULF and short filaments on its own [20,23]. Moreover, zinc-induced vimentin oligomerization precludes subsequent filament formation upon the addition of NaCl, of both wt and C328S vimentin. In contrast, filaments formed by millimolar magnesium are potentiated by the addition of NaCl [20]. However, it has been also found that magnesium or other divalent cations can delay the elongation step of vimentin polymerization, likely through the interaction of numerous magnesium ions with the region of the tail, impeding the head-to-tail interaction necessary for elongation. In addition, magnesium could promote intertetrameric interactions that could slow down the elongation process [20]. Thus, the possibility exists that zinc induces crosslinking at other sites of the protein with different consequences for vimentin assembly.

Of note, under the conditions of the light scattering assay, micromolar ZnCl_2_ and millimolar MgCl_2_, but not NaCl, increased the turbidity of the vimentin solution. This could be due to the bigger size of aggregates induced by both divalent salts, which were the only ones optically detectable. 

Importantly, the effect of zinc on vimentin assembly is strikingly different depending on the polymerization state of the protein (schematized in Figure 7). As stated above, zinc apparently induces an atypical oligomerization of wt and C328S soluble vimentin. Conversely, zinc does not disrupt preformed filaments, but alters their morphology in a manner dependent on the presence of the cysteine residue. Thus, at low micromolar concentrations, zinc selectively increases the diameter of vimentin wt, but not C328S vimentin filaments. Interestingly, we observed that some oxidants and electrophiles disrupt filament assembly by a mechanism dependent on the presence of the cysteine residue and lead to the increased diameter of preformed filaments, which does not occur in vimentin C328S [4]. Therefore, modulation of the environment of the cysteine residue, either by oxidation or hypothetically by zinc binding, could preclude the interactions needed for elongation or induce loosening of preformed filaments.

Our results showed that zinc also induced bundling of preformed vimentin filaments, which appeared more intense in vimentin wt than in C328S, thus indicating that the presence of the cysteine residue contributed to this effect. Recent in vitro observations indicate that zinc binding could alter the rheological properties of vimentin filaments [21], causing a stiffening of the network at low concentrations, but inducing bundle formation at higher concentrations, which resulted in a softening of the network. In this study, calcium exerted similar effects, although at higher concentrations [21].

Interestingly, vimentin appears to bind zinc with high capacity. In cells, fluorescent probes for protein-bound zinc light-up cytoskeletal structures colocalizing with vimentin bundles [3]. Indeed, given the concentration of negative charge surfaces on cytoskeletal proteins, we and others have proposed that these proteins could act as buffers for counterions [18,39] or as a zinc reservoir in the case of vimentin [3]. Notably, in the cellular milieu, millimolar concentrations of magnesium and other divalent cations are present, as well as many other potential cofactors. Nevertheless, vimentin seems to bind zinc with higher affinity than other divalent cations. We previously observed that preincubation with millimolar magnesium slowed down zinc binding to vimentin [3]. Therefore, the interplay between several cations in their interaction with vimentin could occur. Remarkably, examples of interplay between different cations in the binding and/or regulation of protein function can be found in tubulin, as detailed below, and in the case of certain phosphatases, like protein tyrosine phosphatase 1B, which is activated by magnesium and inhibited by zinc [34].

Zinc modulation of vimentin assembly could have both physiological and pathophysiological implications. Since the interaction of zinc with vimentin filaments results in the formation of bundles, this could contribute to its structural role. We have previously observed that zinc-deficient fibroblasts from a patient with a genetic alteration of zinc transport (acrodermatitis enteropathica) displayed vimentin filaments/bundles narrower than those of cells from a control subject, a condition that could be improved by zinc supplementation [3]. With respect to pathological aspects, sub-retinal pigment epithelial deposits associated with age-related macular degeneration contain both high concentrations of zinc [47,48] and vimentin, among other proteins [49,50,51], which could lead to speculating about a role of zinc and vimentin in the formation or progression of these aggregates. Regarding pathological protein aggregates, micromolar zinc has been reported to induce aggregation of other proteins, including Aβ and Tau in the context of neurodegenerative diseases [31,52,53], factor H in relation to age-related macular degeneration [54], and tubulin, inducing atypical polymerization [55]. In the case of tubulin, zinc has been shown to induce the formation of sheets by mechanisms independent of the physiological polymerization elicited by GTP and magnesium [56]. However, a deficit in zinc availability can favor cysteine tubulin oxidation, causing cellular alterations [57], which reveals a complex interplay of zinc homeostasis and tubulin in cells. Analogously, zinc availability appears to modulate the susceptibility of the vimentin network to disruption by oxidants, which could occur by direct and indirect mechanisms [3].

## 4. Materials and Methods 

Materials: Amicon ultrafiltration devices (10 K cut-off) were from Millipore. Slide-A-Lyzer MINI Dialysis devices (20 K cut-off) were from Thermo. PD-SpinTrap G-25 columns were from GE Healthcare. Ninety-six well clear plates were from Falcon. Carbon support grids, MESH CF 400 CU UL, used for electron microscopy were from Aname. High precision cells made of Quartz SUPRASIL, 10 mm path length, were purchased from Hellma. Sypro-Ruby for total protein staining was from Bio-Rad. The polar extract of *E. coli* phospholipids was from Avanti Polar Lipids. Human vimentin wt and C328S, expressed in *E. coli* and purified by established protocols [58], was obtained from Biomedal S.L. (Spain). Zinquin acid and N, N, N’, N’ tetrakis (2-pyridylmethyl) ethylenediamine (TPEN) were from Sigma. Alexa Fluor 488 carboxylic acid succinimidyl ester dye was from Molecular Probes, Invitrogen. High purity salts and other reagents were from Sigma and Merck.

Vimentin refolding and EDTA removal: This was achieved by ultrafiltration followed by dialysis, as previously described [41]. Briefly, vimentin preparations, in 8 M urea, 5 mM Tris-HCl, pH 7.6, 1 mM EDTA, 10 mM β-mercaptoethanol, 0.4 mM PMSF, and approximately 150 mM KCl, were first diluted with EDTA-free buffer and ultrafiltrated using Millipore Amicon filter units (10 K pore size). Ultrafiltrated protein preparations were dialyzed step-wise against 5 mM PIPES-Na, pH 7.0, 1 mM DTT, containing decreasing urea concentrations (6 M, 4 M, 2 M, and no urea), followed by two additional steps against 5 mM PIPES-Na, pH 7.0, 0.25 mM DTT, the last one for 16 h at 16 °C. Finally, proteins were centrifuged at 120,000× *g* for 15 min at 4 °C, discarding the pellet, and aliquots were stored at −80 °C. The protein concentration was estimated from its A280 nm, using an extinction coefficient of 22,450 M^−1^cm^−1^. The final DTT concentration in assays was kept below 0.2 mM to minimize chelating effects [43].

Polymerization assays: Vimentin wt and C328S, at 3.8 µM in 5 mM PIPES-Na, pH 7.0, 0.1 mM DTT, were incubated in the presence of 150 mM NaCl for 15 min at 37 °C or with the indicated concentrations of ZnCl_2_ or MgCl_2_ for 1 h at room temperature (r.t.). In all situations, soluble (S, supernatant) and polymerized (P, pellet) fractions were separated by ultracentrifugation at 55,000 rpm for 30 min at 4 °C. Aliquots of both fractions were resuspended in Laemmli buffer and analyzed by SDS-PAGE, followed by total protein staining with Sypro-Ruby and ultraviolet (UV) light detection.

Turbidity assays: Solutions of vimentin wt and C328S, at 5 μM in 5 mM PIPES-Na, pH 7.0, were placed in Hellma absorption cuvettes, and increasing amounts of ZnCl_2_, MgCl_2_, or NaCl were sequentially added to achieve the indicated final concentrations. Turbidity of the incubation mixtures was immediately monitored at 350 nm, at r.t., using an Agilent Cary 60 UV-Vis spectrophotometer. In the graphs, the absorbance values corresponding to the protein alone were subtracted and were corrected according to vimentin concentration and the volume of the mixtures. Time-dependent measurements were performed every 2 min, for a total of 26 min in the analysis of the reversibility of the effect of ZnCl_2_ and for 16 min in the monitoring of the effect of ZnCl_2_ incubation before or after the addition of NaCl. Turbidity of vimentin solutions without added salts and of incubation buffer with increasing concentrations of salts were monitored, and no changes in the turbidity were detected within the time frame of the experiments.

Protein labeling: Vimentin wt and C328S were covalently labeled in the amino groups by incubation with a three-fold molar excess of Alexa Fluor 488 carboxylic acid succinimidyl ester dye, after which, mixtures were subjected to gel filtration, using PD Minitrap G-25 columns, in order to eliminate the free dye. The extent of vimentin labeling was estimated from the molar absorption coefficients of 22,450 M^−1^cm^−1^ and of 71,000 M^−1^cm^−1^, at 280 nm for vimentin and at 495 nm for the dye, respectively, and ranged between 0.2 and 0.9 moles of fluorophore per mole of protein. Aliquots of vimentin were stored at −80°C, until assayed.

Fluorescence anisotropy: Mixtures of unlabeled vimentin (1 μM) with a tracer amount of Alexa 488-labeled vimentin (50 nM) were incubated with different concentrations of ZnCl_2_, in a Corning 96 well black microplate. Fluorescence anisotropy was determined using a BMG POLARstar Galaxy plate reader, using 485 nm and 520 nm excitation and emission filters, respectively, at 26 °C.

Observation of vimentin assemblies induced by zinc on different supports: For the observation of vimentin assemblies formed in solution, 80 μL aliquots of vimentin, at 5 μM in 5 mM PIPES-Na (pH 7.0), 0.2 mM DTT, were dispersed in a flat-bottomed microplate. Different concentrations of ZnCl_2_ were added to the protein solutions, and the structures formed after 20 min of incubation were visualized under a Zeiss optical inverted microscope coupled to an Axiocam ERc 5s camera.

For visualization of vimentin in microdroplets stabilized by lipids, *Escherichia coli* phospholipids dissolved in chloroform were dried under a nitrogen stream, and the resulting lipid film was resuspended in mineral oil to the final concentration by two cycles of vigorous vortexing and sonication in a water bath. Mixtures of unlabeled vimentin wt (4 μM) and vimentin wt labeled with Alexa 488 (1 μM) were incubated with vehicle or ZnCl_2_ at the indicated concentrations. Next, incubation mixtures were vigorously mixed with the *E. coli* lipids dispersed in mineral oil, and the resulting emulsions were applied on a glass coverslip and observed on a Leica SP2 confocal microscope.

For observation on glass slides, a 5 μL aliquot of vimentin wt or C328S at 5 μM was applied on a glass slide, and 5 μL of vehicle, 200 μM, or 2.5 mM ZnCl_2_ or MgCl_2_ (final concentrations) were added and gently mixed with the pipette tip. Next, a coverslip was placed on top of the mixtures, and samples were visualized on a Leica SP5 confocal microscope using the bright field mode for image acquisition. For zinquin staining, zinquin acid was added to the mixtures at 0.6 mM (final concentration). Fluorescence was detected by excitation with an ultraviolet laser at 405 nm, and emission between 450 and 520 nm was acquired.

Electron microscopy: Soluble oligomeric species of vimentin at 0.2 mg/mL in 5 mM PIPES-Na, pH 7.0 (hypotonic buffer), in the presence of 0.1 mM DTT, were fixed with 0.1% (*v/v*) glutaraldehyde, final concentration. Vimentin filaments were obtained by inducing polymerization with 150 mM NaCl, at 37 °C. Time-lapse monitoring of vimentin assembly was performed by fixing the mixtures at 5, 10, 60, and 120 min after initiating the polymerization. The effect of divalent cations on vimentin assembly or preformed filaments was assessed by incubation with vehicle or the indicated concentrations of ZnCl_2_, for 1 h at r.t, before or after inducing vimentin polymerization by incubation with 150 mM NaCl, for 1 h at 37 °C. Experiments were carried out at least three times, and all incubations were performed in duplicate. Carbon support grids (MESH CF 400 CU UL, Aname) were laid onto drops of glutaraldehyde-fixed mixtures, subsequently washed with water, and negatively stained with 2% (*w/v*) uranyl acetate (Merck). Samples were observed in a JEOL JEM-1230 electron microscope operating at 100 kV, equipped with a CMOS TVIPS TemCam-F416 digital camera. EM images were obtained at 50 K magnification and were processed with FIJI software for the measurement of the width and length of vimentin oligomeric species and filaments. The “plot profile” plugin of straightened filaments was used to determine filament width. On average, 50 vimentin species were measured per experimental condition.

Statistical analysis: All assays were repeated at least three times. The results are presented as the mean values ± standard error of the mean (SEM) or mean values ± standard deviation (SD), as indicated. Statistical differences were evaluated by Student’s t-test and were considered significant for *p* < 0.05. The statistically significant differences are indicated on the graphs. 

## 5. Conclusions

In summary, our observations indicate that zinc exerts complex effects on the organization of vimentin, depending on the polymerization state of the protein and on the presence of the cysteine residue. These differential effects could have implications in vimentin bundling or aggregation in physiological or pathophysiological conditions.

## Figures and Tables

**Figure 1 ijms-21-02426-f001:**
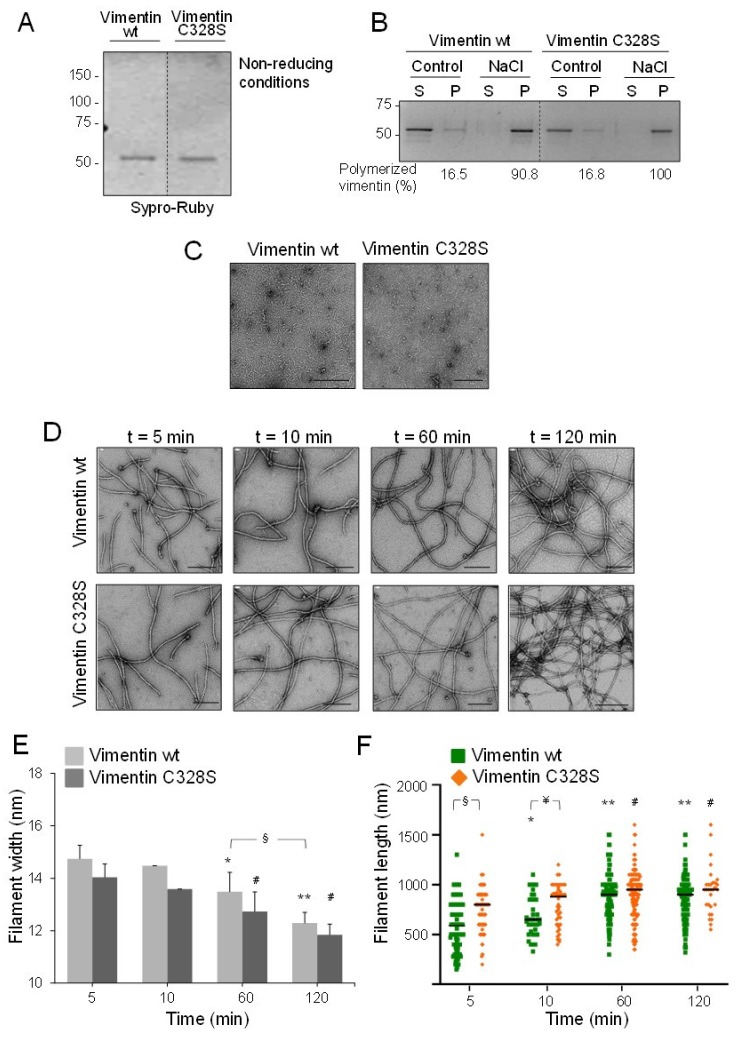
Polymerization of vimentin wt and C328S. (**A**) Vimentin purified and subjected to EDTA removal was analyzed by gel electrophoresis under non-reducing conditions. Total protein was stained with Sypro-Ruby. The position of molecular weight markers is indicated on the left. (**B**) Vimentin wt or C328S in 5 mM PIPES-Na pH 7.0, 0.2 mM DTT was incubated with 150 mM NaCl for 15 min at 37 °C and polymerization assessed by ultracentrifugation. Supernatant (S) and pellet (P) fractions, containing soluble and polymerized vimentin, respectively, were analyzed by SDS-PAGE and Sypro-Ruby staining. The amount of vimentin in each fraction was estimated by image scanning, and the percentage of polymerized vimentin is shown below the gel. Results are representative of five assays. (**C**) Soluble vimentin wt and C328S preparations were analyzed by electron microscopy (EM). Images are representative of four and 10 preparations of vimentin wt and C328S, respectively. (**D**) Vimentin assembly was induced by the addition of 150 mM NaCl at 37 °C, and filament formation at the indicated times was monitored by EM. Images are representative of at least three assays. Images were analyzed with FIJI to determine filament width and length under the different conditions for both vimentin preparations, and the results are shown in (**E**) and (**F**), respectively. Results are average values ± SEM of 50 to 200 filaments per condition. In (**E**), * *p* < 0.05, and ** *p* < 0.001 vs. vimentin wt at 5 min, # *p* < 0.05 vs. vimentin C328S at 5 min, § *p* < 0.05, as indicated. In (**F**), * *p* < 0.05, and ** *p* < 0.0001 vs. vimentin wt at 5 min, # *p* < 0.05 vs. vimentin C328S at 5 min, § *p* < 0.0001, ¥ *p* < 0.05, as indicated, by the unpaired Student’s *t*-test. Scale bars, 200 nm.

**Figure 2 ijms-21-02426-f002:**
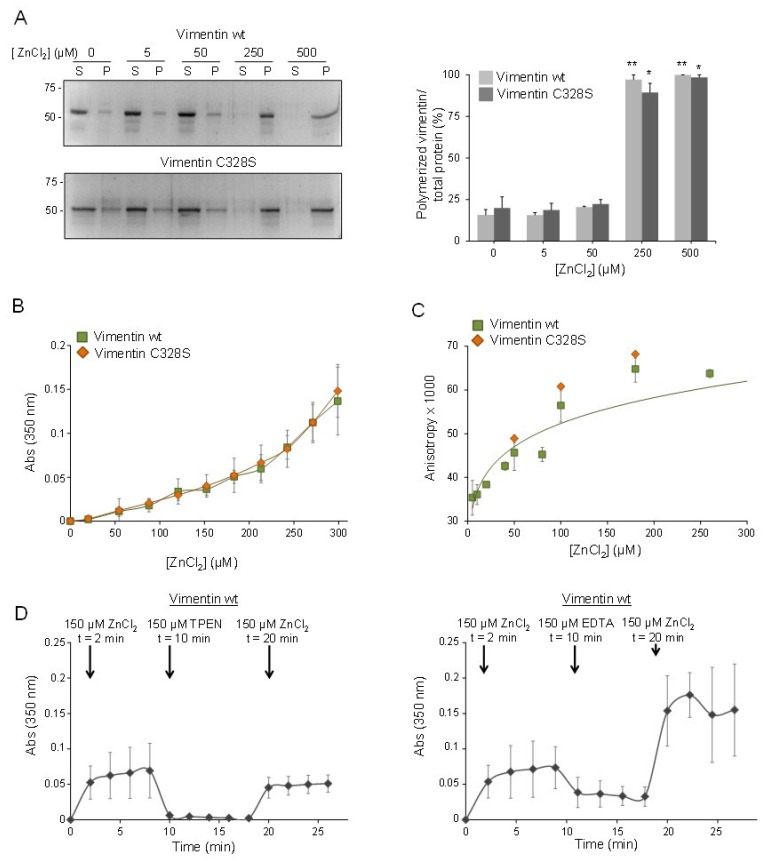
Effect of micromolar zinc concentrations on vimentin oligomerization assessed by centrifugation and turbidity assays. (**A**) Vimentin wt or C328S was incubated with the indicated concentrations of ZnCl_2_ for 1 h at r.t., and the formation of insoluble oligomers was assessed by ultracentrifugation as above. Gels shown are representative of three assays. The proportion of polymerized vimentin, i.e., present in the pellet fraction, was estimated by image scanning, and results corresponding to the mean ± SEM of three assays are depicted in the graphs. ** *p* < 0.005, and * *p* < 0.05 vs. the corresponding construct in the absence of zinc. (**B**) Vimentin wt or C328S was incubated with increasing concentrations of ZnCl_2_, and the turbidity of the solution, assessed as the absorbance at 350 nm at r.t., was measured immediately. Results are the mean values ± SD of at least three experiments. (**C**) Mixtures containing 1 µM unlabeled vimentin and 50 nM Alexa 488-labeled vimentin were incubated with increasing concentrations of ZnCl_2_, and fluorescence anisotropy was monitored at 26 °C. Results are the mean ± SD of eight (vimentin wt) or two replicates (vimentin C328S). (**D**) Reversibility of zinc-induced vimentin aggregation upon addition of TPEN (left) or EDTA (right). Vimentin was sequentially incubated with ZnCl_2_, TPEN, or EDTA and, then, ZnCl_2_ again at r.t., at the concentrations and time-points indicated. Results are average values from three independent experiments. Lines in the graphs are only meant to guide the eye.

**Figure 3 ijms-21-02426-f003:**
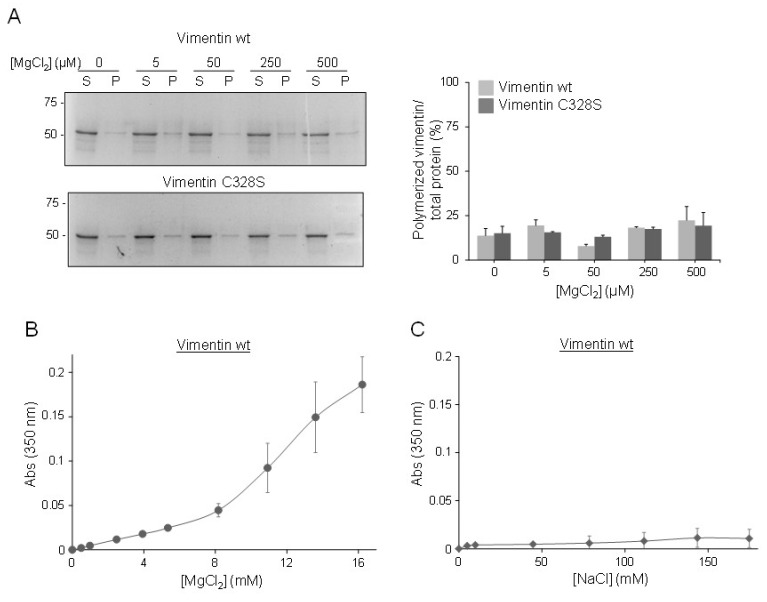
Effect of MgCl_2_ or NaCl on vimentin polymerization. (**A**) Vimentin wt or C328S in 5 mM PIPES-Na pH 7.0, 0.2 mM DTT, was incubated with the indicated concentrations of MgCl_2_ for 1 h at r.t., and polymerization was assessed by ultracentrifugation as above. Supernatant (S) and pellet (P) fractions, containing soluble and polymerized vimentin, respectively, were analyzed by SDS-PAGE and staining with Sypro-Ruby. The percentage of polymerized vimentin for every condition is shown in the graph as average values ± SEM from two assays. (**B**,**C**) Vimentin wt was incubated with increasing concentrations of MgCl_2_ (**B**) or NaCl (**C**) and turbidity measured at 350 nm at r.t. Results are the mean ± SD of three (**B**) or two (**C**) assays.

**Figure 4 ijms-21-02426-f004:**
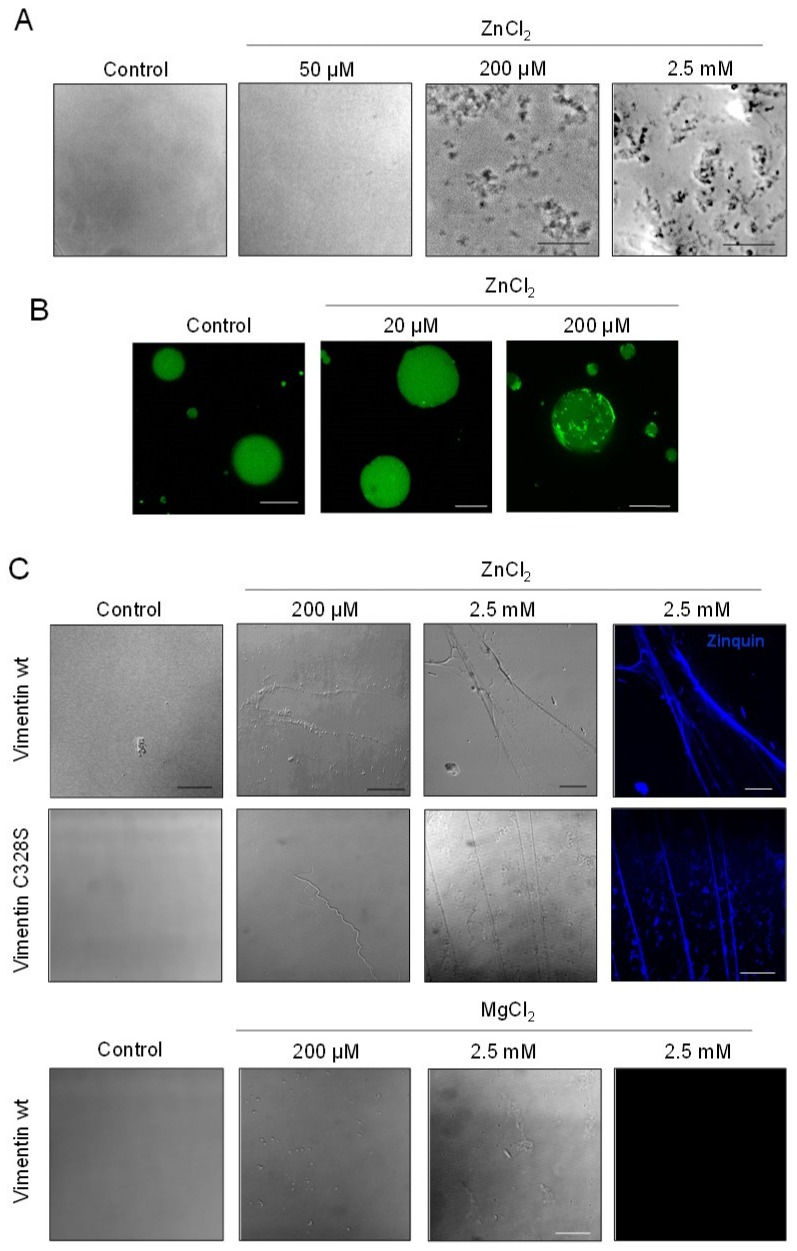
Observation of zinc-induced vimentin structures by optical microscopy. (**A**) Vimentin wt in 5 mM PIPES-Na pH 7.0, 0.2 mM DTT, was incubated with the indicated concentrations of ZnCl_2_ in a plastic microwell plate. Mixtures were observed after 20 min incubation at r.t. Images are representative of three assays. Scale bar, 20 µm. (**B**) Vimentin mixtures containing unlabeled and Alexa 488-labeled vimentin (4 and 1 µM, respectively) were encapsulated in microdroplets in the presence of the indicated concentrations of ZnCl_2_, at r.t., and observed by confocal fluorescence microscopy. Overlay projections, representative of three assays, are shown. (**C**) Vimentin wt or C328S at 5 µM in hypotonic buffer (5 mM PIPES-Na pH 7.0, 0.2 mM DTT) was deposited on a glass coverslip, and the control image was obtained immediately thereafter. Subsequently, ZnCl_2_ or MgCl_2_ was added at the indicated final concentrations, and bright field images were obtained within 10 min of addition at r.t. Finally, zinquin was added at 0.6 mM, final concentration, and structures were visualized by UV detection. Results shown are representative of at least three independent assays with similar results. Scale bars, 20 µm.

**Figure 5 ijms-21-02426-f005:**
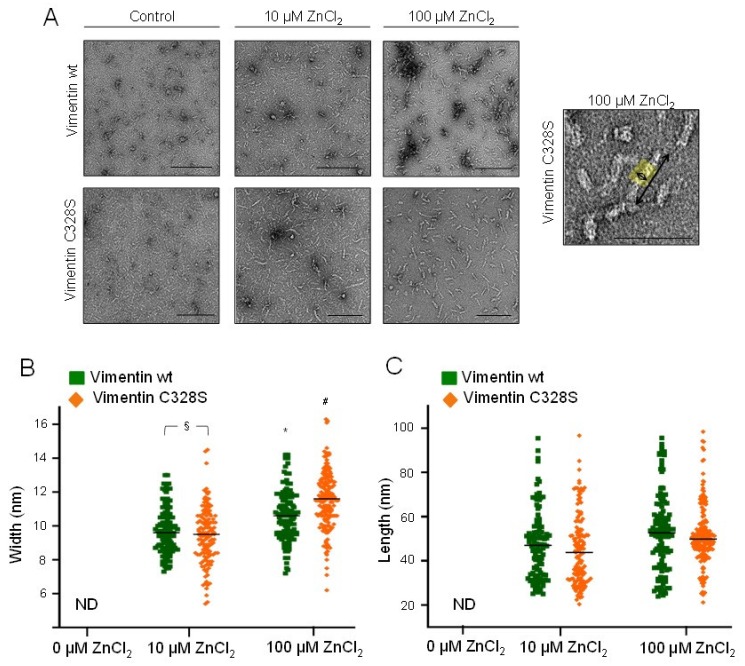
Observation of zinc-induced vimentin structures by electron microscopy. (**A**) Vimentin wt or C328S was incubated with the indicated concentrations of ZnCl_2_ for 1 h at r.t., fixed, negatively stained, and inspected by EM. Scale bars, 200 nm and 100 nm for the right panel. The width and length of the observed fibrils were measured using FIJI software as illustrated in the right panel, and results are summarized in the graphs. (**B**) Width of the structures of at least 20 nm in length. (**C**) Length of the structures observed. Results shown are the mean values ± SEM of 50 “structures” per condition. § *p* < 0.05 vimentin wt vs. C328S, as indicated; * *p* < 0.05 vs. vimentin wt with 10 µM ZnCl_2_; # *p* < 0.05 vs. vimentin C328S with 10 µM ZnCl_2_ by the unpaired Student’s *t*-test.

**Figure 6 ijms-21-02426-f006:**
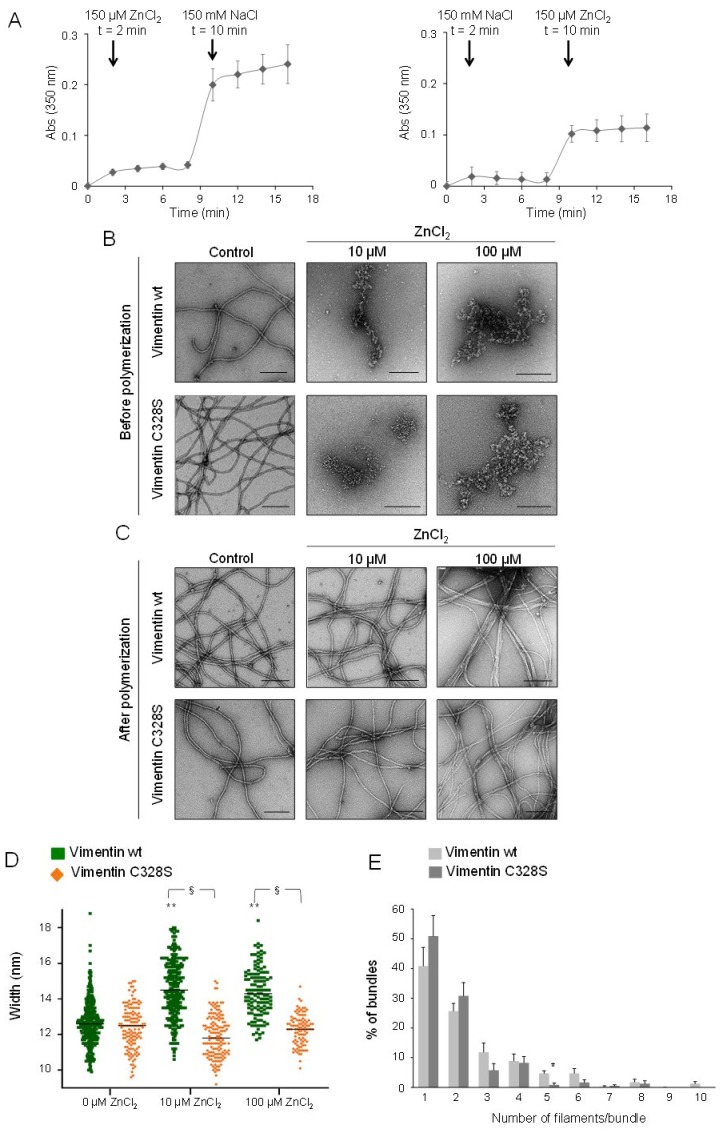
Impact of zinc on the assembly of soluble vimentin and on the morphology of preformed filaments. (**A**) The effect of ZnCl_2_ and NaCl added in different orders, as indicted, on the turbidity of vimentin solutions was monitored at r.t. Results are the average values ± SD of three assays with similar results. (**B**) Effect of the preincubation of vimentin wt or C328S with vehicle or ZnCl_2_ at the indicated concentrations for 1 h at r.t. on subsequent polymerization by the addition of 150 mM NaCl and incubation for an additional hour at 37 °C. (**C**) Vimentin wt or C328S was first polymerized by incubation with 150 mM NaCl for 1 h at 37 °C, after which, ZnCl_2_ at the indicated final concentrations was added, and incubation was continued for 1 h at r.t. The morphology of vimentin wt or C328S filaments was assessed by EM. The images shown are representative of at least three independent assays with similar results. Scale bars, 200 nm. (**D**) The width of filaments shown in (**C**) was measured with FIJI and represented as a scatter plot. Results are the mean values ± SEM of 50 filaments per experimental condition (** *p* < 0.01 vs. wt-control; § *p* < 0.05 vimentin wt vs. C328S by Student’s *t*-test). (**E**) Images from preformed filaments of vimentin wt or C328S treated with 100 µM ZnCl_2_ were inspected for filaments appearing as individual entities or forming laterally associated groups or bundles. Then, the percentage of bundles containing the indicated number of filaments was calculated. At least 230 bundles were counted per experimental condition. Results are the average values ± SEM of four and three assays for vimentin wt and C328S, respectively (* *p* < 0.02 vs. wt by Student’s *t*-test).

**Figure 7 ijms-21-02426-f007:**
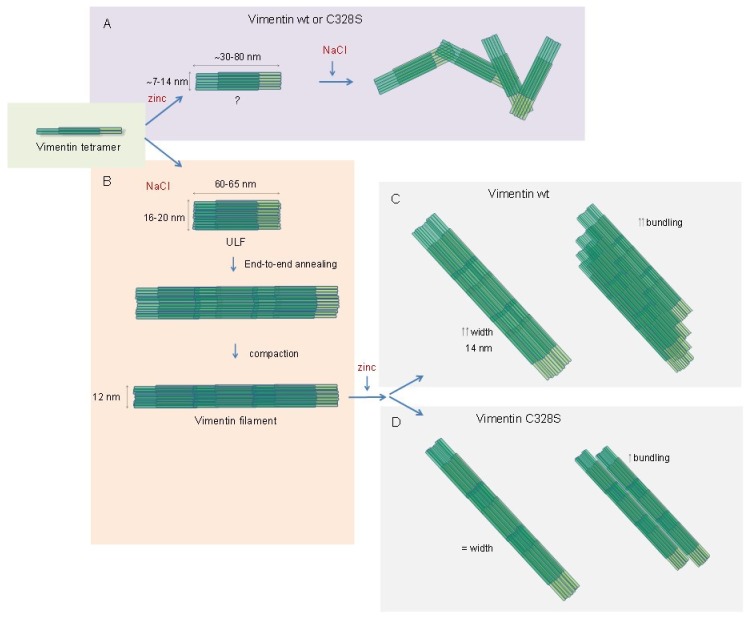
Scheme summarizing the impact of zinc on vimentin assembly as monitored by electron microscopy. Soluble vimentin, represented by a vimentin tetramer, forms atypical structures when incubated with ZnCl_2_ (**A**). Subsequent addition of NaCl does not produce filaments, but entangled aggregates. This effect is similar for vimentin wt and C328S. Conversely, incubation of soluble vimentin with NaCl (**B**) elicits the typical polymerization process, characterized by the formation of unit length filaments (ULF) and end-to-end annealing of these structures that are further compacted to form mature filaments. Incubation of these preformed filaments with zinc leads to a marked increase in width and intense bundling in the case of vimentin wt (**C**), whereas C328S mutant filaments do not suffer changes in width and undergo less intense bundling (**D**).

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
