# Peer review of "Zinc Differentially Modulates the Assembly of Soluble and Polymerized Vimentin"

_ijms, 2020, doi:10.3390/ijms21072426_

Round 1

Reviewer 1 Report

The manuscript by Monico et al. reports results indicating that zinc is  an important factor to modulate assembly of both soluble and polymerized vimentin. In particular, the authors investigate the effects of zinc on vimentin properties. The experiments are well done with appropriate controls and the data obtained support the conclusions. Thus I believe that this manuscript deserves publication in International Journal of Molecular Sciences

Author Response

We thank the reviewer for the kind comments.

Reviewer 2 Report

The manuscript characterizes the assembly properties of vimentin and of a C328S vimentin mutant in vitro, in the presence of zinc and of other cations. It is described that zinc promotes the formation of abnormal assembly products from soluble vimentin, but that addition of zinc to pre-formed vimentin filaments increases slightly their diameter and favors the formation of bundles of multiple filaments. The C328S mutant is less sensitive to these effects on diameter and bundling, suggesting a role of zinc on vimentin regulation via C328.

Overall, the majority of the work in this manuscript is carried out correctly, and the interpretation of the data is careful. Since the present study is restricted to in-vitro-analysis, the physiological effects of zinc on vimentin in a living cell remain largely speculative.

I have two points of criticism that should be addressed in a revised manuscript:

1) Using turbidity measurements, it is described in Figure 3C and in line 216 that NaCl-concentrations up to 150mM don't increase the polymerization of vimentin. How can these results be reconciled with the findings in Figure 1B, where the rapid polymerization of vimentin into filaments is stimulated by 150mM NaCl?

2) The experiment in Figure 6E should be documented correctly: apparently, three independent assays were carried out to investigate the effect of zinc on filament bundling. Graphs in Figure 6E should be established from the data of all three experiments, with the standard deviations indicated in the graphs. Eventually, this experiment may require re-counting, to be fully convincing.
